# Time-resolved in situ visualization of the structural response of zeolites during catalysis

Jinback Kang[1], Jerome Carnis [1], Dongjin Kim[1], Myungwoo Chung[1], Jaeseung Kim[1], Kyuseok Yun[1], Gukil An[1], Wonsuk Cha [2,12], Ross Harder [3], Sanghoon Song [4], Marcin Sikorski[4], Aymeric Robert[4], Nguyen Huu Thanh[5], Heeju Lee[1,6], Yong Nam Choi[6], Xiaojing Huang[7], Yong S. Chu[7], Jesse N. Clark[8,9], Mee Kyung Song[5], Kyung Byung Yoon[5], Ian K. Robinson [10,11] & Hyunjung Kim [1✉]

Zeolites are three-dimensional aluminosilicates having unique properties from the size and connectivity of their sub-nanometer pores, the Si/Al ratio of the anionic framework, and the charge-balancing cations. The inhomogeneous distribution of the cations affects their catalytic performances because it influences the intra-crystalline diffusion rates of the reactants and products. However, the structural deformation regarding inhomogeneous active regions during the catalysis is not yet observed by conventional analytical tools. Here we employ in situ X-ray free electron laser-based time-resolved coherent X-ray diffraction imaging to investigate the internal deformations originating from the inhomogeneous Cu ion distributions in Cu-exchanged ZSM-5 zeolite crystals during the deoxygenation of nitrogen oxides with propene. We show that the interactions between the reactants and the active sites lead to an unusual strain distribution, confirmed by density functional theory simulations. These observations provide insights into the role of structural inhomogeneity in zeolites during catalysis and will assist the future design of zeolites for their applications.

---

[1] Department of Physics, Sogang University, Seoul 04107, Korea. [2] Materials Science Division, Argonne National Laboratory, Argonne, IL 60439, USA. [3] Advanced Photon Source, Argonne National Laboratory, Argonne, IL 60439, USA. [4] Linac Coherent Light Source, SLAC National Accelerator Laboratory, Menlo Park, CA 94025, USA. [5] Department of Chemistry, Sogang University, Seoul 04107, Korea. [6] Korea Atomic Energy Research Institute, Daejeon 34057, Korea. [7] National Synchrotron Light Source II (NSLS-II), Brookhaven National Laboratory, Upton, NY 11973, USA. [8] Stanford PULSE Institute, SLAC National Accelerator Laboratory, Menlo Park, CA 94025, USA. [9] Center for Free-Electron Laser Science, Deutsches Elektronensynchrotron (DESY), 22607 Hamburg, Germany. [10] London Centre for Nanotechnology, University College London, WC1E 6BT London, UK. [11] Condensed Matter Physics and Materials, Brookhaven National Laboratory, Upton, NY 11973, USA. [12] Present address: Advanced Photon Source, Argonne National Laboratory, Argonne, IL 60439, USA. ✉email: hkim@sogang.ac.kr

Zeolites form an important class of materials with frameworks consisting of Si, Al, and O atoms. Since the framework is negatively charged owing to the presence of $(AlO_4)^-$ sites, charge balancing cations should exist within the void spaces. The chemical nature of the framework, the number and type of cations that balance the negative charges of the frameworks can be tuned. Zeolites have been extensively used as catalysts for various reactions, including refinement of crude oil[1] and reduction of nitrogen oxides ($NO_x$) in vehicle exhausts[2], adsorbents for various molecules including carbon dioxide[3], cation exchangers[4], size-selective separation of molecules[5], and many others[6]. These unique properties of zeolites arise from their sub-nanometer scale pores, which vary in size and shape depending on the type of zeolites. Because charge-balancing cations exist near the Al centres, the amount of charge-balancing cations decreases when the Si/Al ratio of the framework increases[7,8]. Typically, in an MFI type zeolite like ZSM-5, the Si/Al ratio is maximum at the core and gradually decreases from the core to the external surface[7]. Therefore the density of charge-balancing cation is the lowest at the core and the highest at the surface[9]. Besides, if zeolites are made using an organic structure-forming template, then the core of the crystal is likely to have a higher degree of organic residue[10], which might induce inhomogeneities in cation density in a crystal. Accordingly, since the pores are extremely small, the diffusion rates of molecules within the pores are greatly affected by the number of cations, defects, and residues within the pores. There are the inhomogeneous regions where molecules are more or less readily adsorbed.

Indeed, it has been known that the adsorbed molecules are not evenly distributed within the zeolite crystals[11]. In other words, their inhomogeneous distributions within crystals sensitively affect their performances because these factors influence the intra-crystalline diffusion rates of the reactants and products[12]. Therefore time-resolved in situ visualization of the active regions in zeolite crystals during the chemical process and the subsequent elucidation of the factors within zeolite crystals provide important information to maximize the usage of zeolite as catalysts and to design and synthesize zeolite catalysts with enhanced performances[13,14].

However, such local information cannot be obtained by the conventional techniques such as nuclear magnetic resonance[15], infrared absorption spectroscopy[16], X-ray powder diffraction[17], and X-ray absorption spectroscopy[18], because these methods only provide information regarding the average properties of a crystal. This shortcoming is addressed in the current work by employing a spatially sensitive strain imaging technique.

Coherent X-ray diffraction imaging (CDI) involves the computational reconstruction of real-space images from coherent X-ray scattering patterns using phase retrieval algorithms[19]. Bragg geometry CDI (BCDI) allows obtaining the shape of an object as well as the displacements and strains[20,21] (see Supplementary Note 1). Time-resolved BCDI measurements with X-ray free electron lasers (XFELs) provide kinetic information of the internal displacement distribution, with ≥100 times higher time resolution than synchrotron sources due to their intense and fully transversely coherent X-ray beam[22].

Catalytic deoxygenation of $NO_x$ into $N_2$ is an important reaction to reduce the emission of harmful exhaust gas into the atmosphere. In practice, Cu(II)-exchanged SSZ-13 and SAPO-34 are currently widely used as effective catalysts for $NO_x$ deoxygenation[23,24]. However, the crystal quality of those crystals is not yet reached for very sensitive detection using coherent X-ray diffraction imaging. Therefore we used Cu(II)-exchanged ZSM-5 as an example of zeolite catalyst for deoxygenation of $NO_x$[25] to $N_2$ with propene as the reducing agent[26]. Because this zeolite was originally developed for the above reaction and found to be most appropriate for showing sensitive changes with different catalytic molecules.

Here, we show how the inhomogeneities affect the crystal strain in Cu(II)-exchanged ZSM-5 during the catalytic process of deoxygenation of $NO_x$ to $N_2$ with propene as the reducing agent by employing time-resolved in situ BCDI measurements with XFELs. We observe an unusual displacement field distribution due to the interactions between the reactants and the inhomogeneous active sites within the Cu-ZSM-5 crystal, supported by density functional theory simulation and finite element analysis.

## Results

**In situ deformation field evolution during the catalytic process.** A schematic of the in situ time-resolved BCDI experiment on the (200) Bragg reflection from a Cu-ZSM-5 microcrystal is shown in Fig. 1a. Two dimensional (2D) Coherent X-ray diffraction (CXD) patterns[27] were recorded during the entire catalytic process. The initial parameters for the phase retrieval process were set from the 3D CXD data for the crystal under $N_2$ (Supplementary Fig. 1). To observe structural responses with different reactants, our study was conducted in stages; process (i) for propene adsorption (Fig. 1b; in blue) and process (ii) for $NO_x$ deoxygenation (in red) by inserting NO and $O_2$ in the presence of adsorbed propenes. The reaction temperature of the deoxygenation of $NO_x$ is typical ~500 °C but can be reduced down to 180 °C in the presence of propene[28]. For $NO_x$ deoxygenation reaction used for purification of vehicle exhausts, it is crucial to observe the effect of

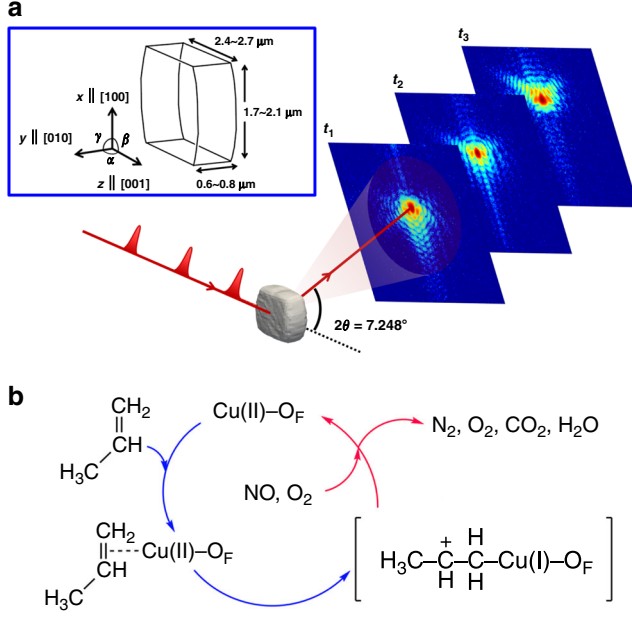

**Fig. 1 Schematic of the experimental principles and reactions.**
**a** Schematic of the time-resolved BCDI experiment at LCLS. The Cu-ZSM-5 is a monoclinic structure, where $\alpha \cong 90.4°$ and $\beta = \gamma = 90°$. **b** Diagram illustrating the catalytic deoxygenation reactions in the Cu-ZSM-5. Two processes are used in this study: (i) propene adsorption (in blue) and (ii) $NO_x$ deoxygenation (in red) by inserting NO and $O_2$ in the presence of adsorbed propenes. In the initial state in $N_2$, Cu(II) ions are coordinated to O atoms ($O_F$) adjacent to Al atoms. During the process (i), the propene double bond coordinates to a Cu(II) ion to form $CH_3CH=CH_2-Cu(II)$[29]. This complex undergoes electron transfer to produce $CH_3CH(+)CH_2-Cu(I)$[28,29]. In process (ii), after a mixture of NO and $O_2$ is inserted, $O_2$ oxidizes the hydrocarbon portion of $CH_3CH(+)CH_2-Cu(I)$, generating $CO_2$ and $H_2O$, while $NO_x$ reacts with Cu(I) to give $N_2$ and $O_2$[28].

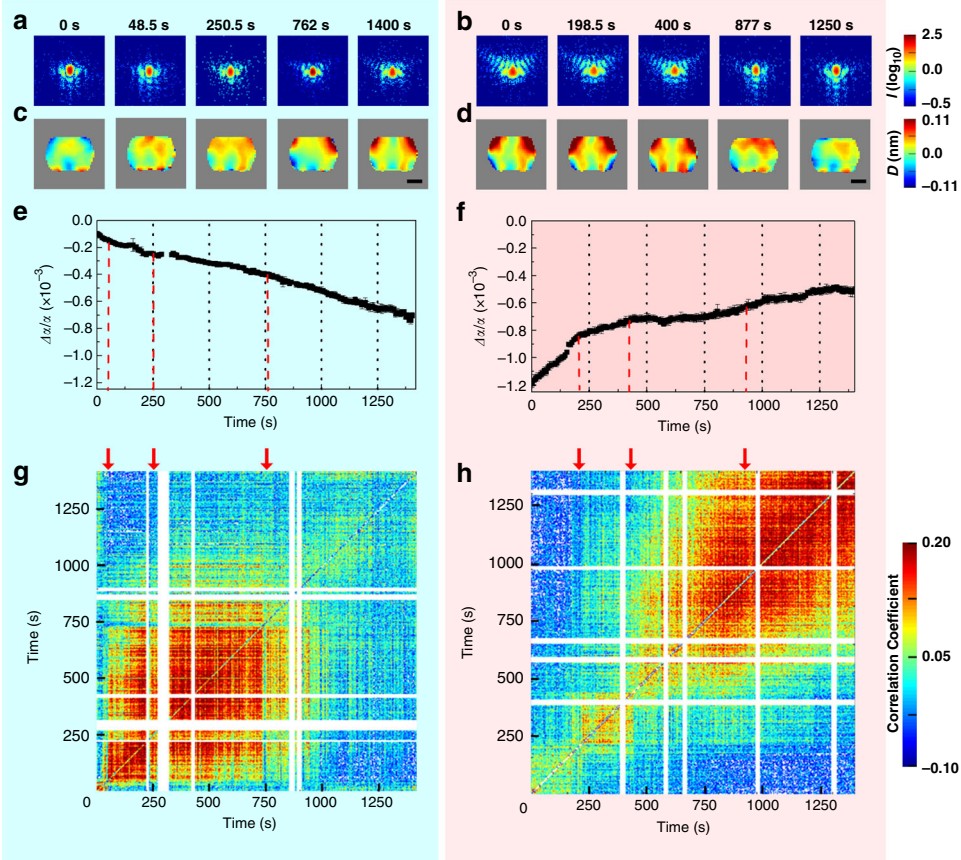

**Fig. 2 Lattice constant and deformation during propene adsorption and NO$_x$ deoxygenation.** The CXD patterns (**a**, **b**), the projected displacement field maps (**c**, **d**), the strain-rate coefficients (**e**, **f**), and the cross-correlation coefficient plot of displacement field results (**g**, **h**) during propene adsorption (in the light blue panel) and NO$_x$ deoxygenation process (in the light magenta panel), respectively. During the propene adsorption, the CXD patterns in **a**. Show that the fringes beside the central peak start rising to a higher wave vector at 250.5 s and continue until 1400 s. Once the NO$_x$ deoxygenation starts, the patterns in **b**. Change significantly but abruptly return to the initial state at 877 s. The projected displacement field during propene adsorption in **c**. Shows an arched shape at 48.5 and 250.5 s, which spreads to the upper side and becomes enhanced at 1400 s. In the NO$_x$ deoxygenation, the displacement in **d**. Shows maximum displacement at the beginning, and then returns to its original state at 877 s. The scale bar corresponds to 1.0 μm. The strain-rate coefficients in **e**, **f** are provided in Supplementary Table 2. The cross-correlation of displacements in **g**, **h** shows coefficients from −0.1 to 0.2, where 1 indicates identical displacement, 0 no relation, and a negative sign the opposite direction. The white lines are due to missing data related to the instability of the XFEL beam.

hydrocarbons on the catalyst structure during the deoxygenation reaction, since hydrocarbons produced by incomplete combustion are inherently present in the exhausts. In this work, we fixed the temperature to 250 °C based on results from Fourier transform infrared spectroscopy (FTIR) (Supplementary Fig. 2) and gas analysis by mass spectrometry (Supplementary Fig. 3). The products identified in each process were in good agreement with previous studies[28–30] (Supplementary Note 2).

Figure 2 presents the results during the two processes (i) and (ii) at a selected time when distinct changes are observed. The 2D CXD patterns shown in Fig. 2a, b consist of the sum of 60 subsequent pulses (corresponding to 0.5 s), as a result of a trade-off optimization process between signal-to-noise ratio and time resolution (Supplementary Note 3). The retrieved projected displacements (Fig. 2c, d) range from 0.11 nm (in red, in the direction parallel to the (200) wavevector) to −0.11 nm (in blue, opposite direction). In process (i), distinct changes appear at 250.5 s and persist until 1400 s. At 1400 s, they reach a maximum value of 0.10 nm at the upper edges, corresponding to ~10% of the (200) lattice spacing.

After remnant propene gases were removed from the sample environment, NO and O$_2$ gases were inserted in the presence of adsorbed propenes. In process (ii), the CXD patterns were found

to change more abruptly, and the displacements at the upper edges of the crystal are ~10% larger than in the process (i). By 1250 s, the entire crystal returns almost back to the original state (i.e., diffraction pattern and displacements) before the reaction started in (i).

To observe the overall lattice changes at a fixed temperature (250 °C) as a function of time, we use a "strain-rate coefficient," $\alpha(t)$, defined as the average lattice constant change per unit time ($\Delta a$) to the lattice constant ($a$) in N$_2$. It shows a negative slope during the process (i) (Fig. 2e) but a positive slope during the process (ii) (Fig. 2f), thus indicating first a contraction followed by an expansion of the crystal lattice. It is clear that the interactions between the Cu ions and the introduced gas molecules are reflected in the overall average variations in the lattice spacing. The timestamps in Fig. 2a, b were selected based on the times where the slope of $\Delta a/a$ changed.

To quantify when displacement changes during the processes (i) and (ii), a Pearson correlation function was applied to the lattice displacements (Fig. 2g, h). A cross-correlation analysis[31] reveals that the crossovers in the correlations coincide with times for which the slopes of $\Delta a/a$ change. However, a similar analysis of the image amplitudes presents less correlation (Supplementary Fig. 4). This behavior can be inferred to the fact that the

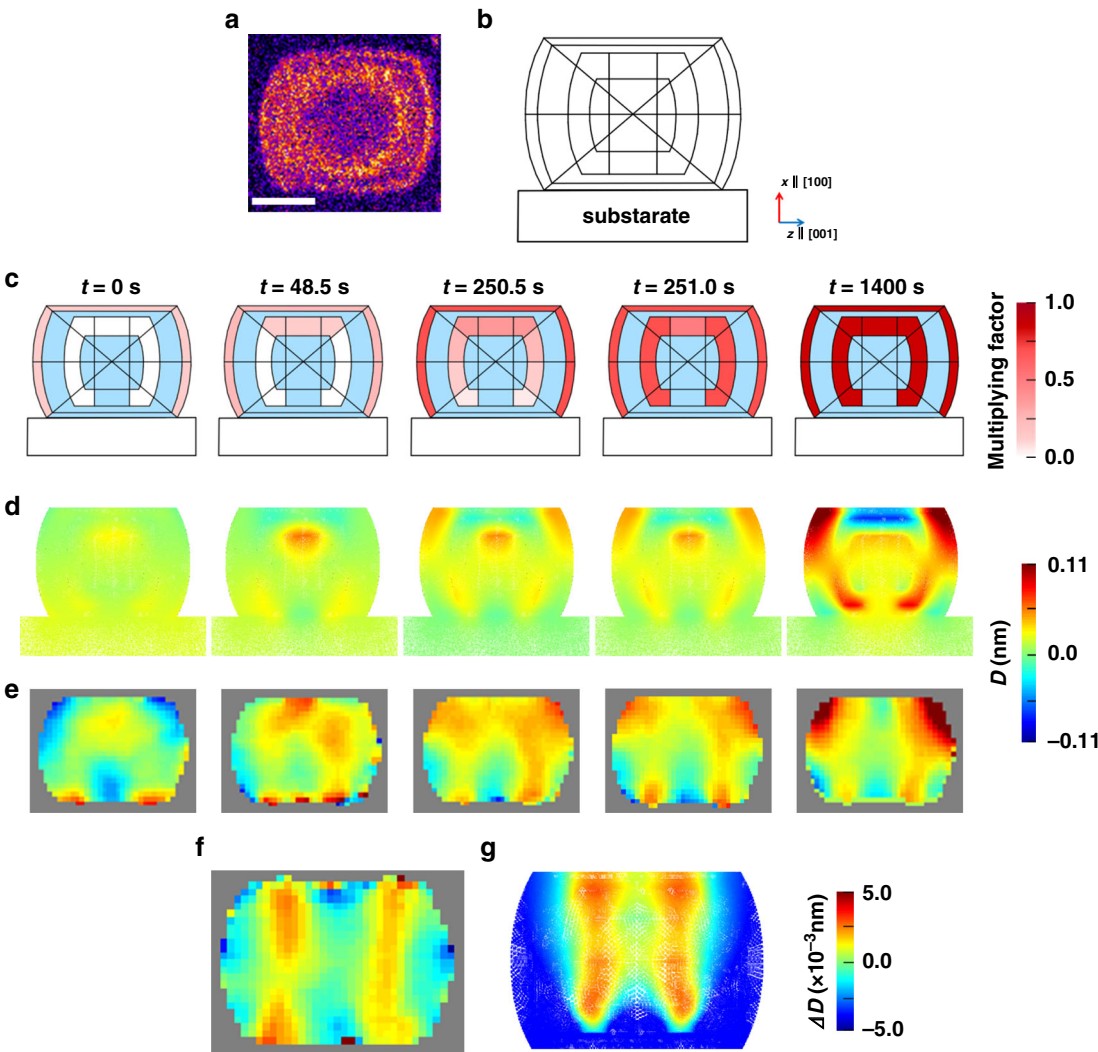

**Fig. 3 X-ray fluorescence microscopy and FEA results during propene adsorption. a** X-ray fluorescence microscopy image of a Cu-ZSM-5, in which the image brightness correlates with Cu density. The scale bar corresponds to 400 nm. **b** Model based on the XFM results. **c** Adsorption states of the propene, **d** FEA results, and **e** the measured projected displacement at $t = 0$, 48.5, 250.5, 251.0, and 1400 s during the propene adsorption process. Fractional values of $\alpha(t)_{(200)}$ and $\alpha(t)_{(020)}$ were used for the red areas in **b**, multiplying by a factor of 0 to 1 to reflect progressive propene adsorption. After $t = 1400$ s, the full coefficient values were used. The differences in the displacement at $t = 250.5$ and 251.0 s from **f** the experimental results and **g** FEA simulation.

amplitude of reconstruction is proportional to the diffracting electron density of the material and is therefore relatively insensitive to the chemical reaction process.

**The inhomogeneity of cation distribution and finite element analysis.** To find the origin of the unusual strain development during the adsorption and catalytic processes, an X-ray fluorescence microscopy (XFM)[32] image of a Cu-ZSM-5 is shown in Fig. 3a. Yellow points indicate large Cu densities with an inhomogeneous distribution, forming ring-like regions. The Cu ions bind near Al sites, which are typically inhomogeneously distributed in ZSM-5 as a result of their synthesis[7]. Note that in XFM, the $xz$ plane of the sample is attached on the substrate whereas in CXD the $yz$ plane does. Using a model based on the XFM results (Fig. 3b), we used finite element analysis (FEA) to calculate the expected displacement field inside the crystal along the (200) direction. We used the strain-rate coefficients, $\alpha(t)_{(200)}$ obtained from Fig. 2e, f for the process (i) and (ii), respectively, and $\alpha(t)_{(020)}$ obtained from another crystal oriented at (020) in the same set of measurements. The coefficients are listed in the Supplementary Table 2. $\alpha(t)_{(002)}$ is assumed to be constant.

Figure 3c–e present the adsorption states of propene, the FEA results, and the experimental data, respectively, at $t = 0$, 48.5, 250.5, 251.0, and 1400 s during the process (i). The degree of adsorption is calculated by multiplying a factor (0 to 1) to the strain coefficient of propene, $\alpha(t)_{(200)}$ and $\alpha(t)_{(020)}$ at $t = 0 \sim 762$ s. The areas in blue represent propene non-absorbed regions; therefore, the values of $\alpha(t)_{(200)}$ and $\alpha(t)_{(020)}$ in vacuum are used (Supplementary Table 2). In our models, the lower part of the central ring is assumed to be a non-absorbed area due to the geometry of the sample used in CXD. At $t = 0$ s, immediately after the introduction of propene, the propene molecules are assumed to adsorb only at the crystal surface. This observation is similar to that of a "core-shell" structure[10] and results in a triangular deformation field. At 48.5 s, propene molecules start to adsorb at the top part of the internal ring inside the crystal. At 250.5 and 251.0 s, where $\Delta a/a$ is in the plateau in Fig. 2e, there is an alternating displacement field patterns in the form of connected columns in arched shapes. The displacement fields in the extended time range from 249 to 253 s in Supplementary Fig. 5 show similar alternating patterns at 250.5–251.0 s and 252.5–253.0 s. We selected the data at 250.5 and 251.0 s for

demonstrating the detailed process of adsorption using the strain analysis. It is interpreted that propene molecules are adsorbed at the outer side of the internal ring (Supplementary Fig. 6 and Supplementary Table 3). At $t = 1400$ s, the Cu sites appear to be occupied entirely with propene molecules.

Even though the results at $t = 250.5$ and 251.0 s are very similar, the subtraction between them shows alternating column-like features with a maximum value of $4.0 \times 10^{-3}$ nm (Fig. 3f), and shows a good agreement with simulation (Fig. 3g). This demonstrates the possibility of measuring the details of the propene adsorption process with a 0.5 s time resolution.

**The reactant effects of the deformation field distribution**. If we consider the deformation behavior related to the strain-rate coefficients at a fixed temperature, one might expect a release of the deformation after the insertion of NO and $O_2$. However, the deformation is observed to be more severe than that with propene only. It implies that a simple strain-rate coefficient model cannot fully explain the deformation during the catalytic $NO_x$ deoxygenation. Therefore, we calculated the effects of the reactants on the lattice using density functional theory (DFT). The DFT potentials are calculated for a single ZSM-5 unit cell with a single Cu ion placed next to the Al site in the pore, and the propene molecule situated near the Cu ion during $NO_x$ deoxygenation. The detailed process of the DFT calculation is described in the Methods. The most stable positions of the molecules are shown in Fig. 4a–c for the NO and $O_2$ molecules located at another channel, therefore, separated from the Cu active sites, whereas in Fig. 4e–g those molecules located close to the active sites in the same pore.

We applied the stress tensors ($\sigma$) obtained by DFT to the red area of the model in Fig. 3c at $t = 1400$ s and the FEA calculation results with $\sigma_{far}$ and $\sigma_{close}$ are shown in Fig. 4d, h, respectively. We observe that NO and $O_2$ molecules are located near the propene, and the Cu active sites in the pore, which results in the generation of an unusual strain distribution during the $NO_x$ deoxygenation.

## Discussion

In conclusion, we have presented XFEL measurements of the displacement distribution in a Cu-ZSM-5 crystal during the catalytic $NO_x$ deoxygenation with propene. The primary reason for the development of a strain field during the initial stage of the reaction is attributed to the propene adsorption at inhomogeneously distributed Cu sites. DFT-based FEA calculations verified that additional forces induced by the coordination of reactant gases with Cu ions induce the strain observed by BCDI during the catalytic $NO_x$ deoxygenation. Coherent and intense XFEL pulses provided maps of these structural changes on a time scale not available from conventional synchrotron sources. Our micro-chemical engineering approach opens up new avenues for the atom-by-atom design of nano-catalysts with distinct and tunable chemical activity, specificity, and selectivity.

## Methods

**Sample preparation**. ZSM-5 is a medium pore size aluminosilicate zeolite developed by Mobil Corporation[33]. It has 10-membered partially elliptical O-ring pore/channel systems with opening sizes[34] of $0.54 \times 0.56$ nm$^2$ and $0.51 \times 0.54$ nm$^2$, with a Si/Al molar ratio ranging between 30 and $\infty$ (all Si, no Al).

ZSM-5 crystals are hydrothermally synthesized from a gel consisting of tetraethyl orthosilicate, sodium aluminate (NaAlO$_2$; 35% Na$_2$O, and 35% Al$_2$O$_3$), tetrapropylammonium hydroxide (TPAOH), and potassium hydroxide by heating at 200 °C for 24 h. The resulting ZSM-5 has a Si/Al molar ratio of 44 with formula Si$_{93.85}$Al$_{2.15}$O$_{192}$. The sizes of the ZSM-5 microcrystals used in this study are 1.7–2.1 μm in height, 2.4–2.7 μm in width, and 0.6–0.8 μm in depth.

The ZSM-5 microcrystals were attached to a Si wafer substrate using 0.035 wt.% polyethyleneimine (PEI) (Mw of approximately 1300) as an adhesive and the ZSM-5 samples on the substrate were then calcined at 550 °C for 12 h to remove residual TPAOH and PEI[10]. After calcination, the specimens underwent a Cu(II) exchange

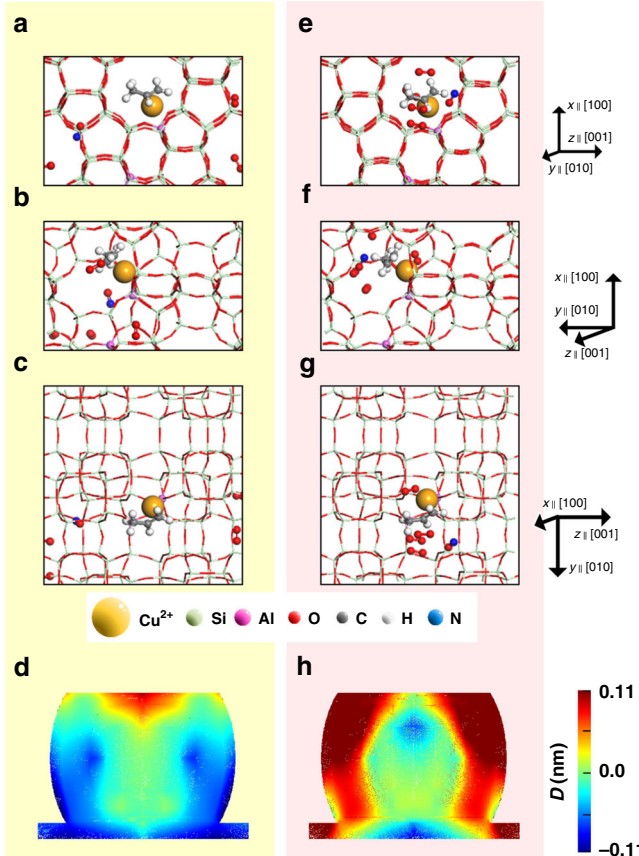

**Fig. 4 Stabilized atomic positions from DFT calculations and respective FEA simulations in $NO_x$ deoxygenation.** The simulated atomic positions are viewed along **a** [010], **b** [001], and **c** [100] direction, when NO and $O_2$ molecules are located in another channel separated from the Cu active sites, and **d** the respective FEA simulation result. When NO and $O_2$ are placed near the active site, the simulated atomic positions are viewed along **e** [010], **f** [001], and **g** [100], and **h** the respective FEA simulation result.

process consisting of immersion in a 0.25 M CuCl$_2$ solution for 24 h. Each ion-exchanged sample was briefly dried in a stream of N$_2$ gas at room temperature and then placed under vacuum at 120 °C for 15 h. The samples were examined by X-ray diffraction (Supplementary Fig. 7 and Supplementary Table 4) before and after the Cu ion exchange process; no effect on the structure was detected. X-ray photoelectron spectroscopy was employed to measure residual Cl element after the Cu ion exchange with CuCl$_2$ (Supplementary Fig. 8). No peak around the chloride position was observed.

**Coherent X-ray diffraction experiments**. Coherent X-ray diffraction measurements were performed at the XCS instrument[35] at the Linac Coherent Light Source (LCLS, SLAC National Accelerator Laboratory, USA)[36]. A double crystal monochromator set the X-ray energy at 8.8 keV via Si (111) monochromator, and compound refractive lenses were used to focus the FEL beam to a spot size of 30 × 30 μm$^2$ (H × V) on the sample. To avoid sample damage by the focused high-intensity FEL beam, attenuators were used with the transmission of $9.71 \times 10^{-2}$. The bandwidth of Si (111) monochromator is $\Delta\lambda/\lambda = 1.36 \times 10^{-4}$, which corresponds to the longitudinal coherence length of 1.1 μm[37]. Since the beam is ~93% transversely coherent[22], the transverse coherence length is ~$9.3 \times 10^4$ μm in both the horizontal and vertical directions. Therefore, the beam used in this study is sufficiently transversely coherent.

The resulting 2D CXD patterns at the (200) Bragg peak ($2\theta = 7.248°$) were collected with the Cornell SLAC hybrid pixel array detector (CSPAD) (375 × 390 pixels with 110 × 110 μm$^2$ pixel size). 3D CXD data were collected using the rocking curve around the Bragg peak with a range of $\Delta\theta = \pm0.25°$ and a 0.01° step size. A total of 51 patterns were acquired for each 3D rocking scan. The 3D image in Supplementary Fig. 1 is retrieved from the coherent diffraction patterns accumulated over 60 shots, which have the same S/N ratio as the 2D images. So-called "dark images" were collected prior to the experimental run cycle for background subtraction. The sample to detector distance was 7.25 m, and an evacuated section was inserted between the sample and the detector to prevent air

scattering. CXD patterns were recorded for each FEL pulse at a repetition rate of 120 Hz. The exposure to each gas was synchronized with the measurement.

**Phase retrieval algorithm.** Phase retrieval algorithm was applied to the 2D and 3D diffraction patterns, using a guided approach integrated with error reduction (ER)[19] and difference mapping (DM)[38] with a phase constraint range of ±π. Five random starts were initiated, with each member being subjected to 180 iterations of the relaxed DM and 20 alternating iterations of the ER algorithm. The total number of iterations for each reconstruction was 2050. A final 50 iterations of the ER algorithm were applied to conclude the reconstruction. The best reconstruction was selected after each generation and applied to produce another 15 new iterates. This procedure was repeated until five generations completed and the final data set was obtained by averaging the five best iterates.

The support of the object in real space was fixed according to the shape obtained from 3D data under $N_2$. The fractional error in the diffraction amplitude was less than 0.040 on average.

**Estimation of spatial resolutions in CDI.** Assuming that a reconstructed image in real space, $g(\mathbf{x})$, is obtained by the phase retrieval process, its Fourier transform is expressed by $\mathcal{F}_w(g(\mathbf{x})) = G(\mathbf{w}) = |G|\exp(i\varphi(\mathbf{w}))$, where $\varphi(\mathbf{w})$ is the retrieved phase and $\mathbf{w}$ is the spatial frequency, in which each spatial frequency can be regarded as a volume grating. The square of the modulus, $|G|^2$, is the measured intensity $I(\mathbf{w})$. Phase retrieval transfer function (PRTF) is defined by

$$\mathrm{PRTF}(\mathbf{w}) = \frac{|G(\mathbf{w})|}{\sqrt{I(\mathbf{w})}} = \frac{||G|\exp(i\varphi(\mathbf{w}))|}{\sqrt{I(\mathbf{w})}}. \tag{1}$$

The resolution cutoff of the phase retrieval process is estimated conservatively as the spatial frequency $\mathbf{w}$ when the PRTF reaches 0.5[39]. Supplementary Fig. 9 shows the plot of PRTF in the present experiment, calculated from the 3D and 2D reconstruction results. The cutoff frequency for the 3D reconstruction is $w = 0.0114 \ \text{Å}^{-1}$, thus the spatial resolution ($l$) of the 3D image is 55.1 nm. For the 2D reconstruction, the cutoff frequencies in $z$- and $x$-direction have the same value, $w_z = w_x = 0.00839 \ \text{Å}^{-1}$, thus the spatial resolution of the 2D images is $l_z = l_x = 74.9 \pm 1.1$ nm. We used ~4700 data sets for the error bars in the resolution of the 2D reconstruction.

**In situ control of the ambient atmosphere with different gases.** Cu-ZSM-5 samples were placed in a chamber with a gas environment remotely controlled. The sample temperature was fixed at 250 °C. The pressure and temperature status of the chamber and the gas injection system were monitored. The environment in the chamber was transitioned from pure $N_2$ to 1% propene in $N_2$ and then to 1% NO with 10% $O_2$ in $N_2$. For exchange gases, the sample chamber was evacuated shortly to remove residual gases before the insertion of the desired gases. The gases were preheated to 250 °C before their introduction to avoid undesirable thermal expansion of the sample. The thermal fluctuation during the gas exchange process was <2 °C.

**Fourier transform infrared spectroscopy.** FTIR spectra were acquired to clarify the intermediate products and mechanisms. The gas exchange process was identical to that during the CXD experiments. However, the concentrations of the gases were different from those used in the CXD measurement to identify the intermediates during the processes, i.e., 100% pure propene and a 2:1 volume mixture of NO and $O_2$. The measurements were conducted in the following steps. The FTIR spectra on Cu-ZSM-5 powders were recorded in vacuum, propene, and NO and $O_2$ gases at RT, 250 °C, and 400 °C. NO and $O_2$ gases are inserted in the presence of the adsorbed propene molecules. Before the measurement at a different temperature, the sample was heated at 400 °C under vacuum to remove any residual molecules inside the sample. Each FTIR spectrum was acquired over the range from 1400 to 4000 cm$^{-1}$.

**Mass spectrometry.** Mass spectrometry (Prisma Plus$^{\mathrm{TM}}$ QMG220) measurements were acquired from Cu-ZSM-5 powders in a quartz tube (6 mm in diameter and 240 mm in length) at 250 °C with He flow (50.0 cm$^3$ min$^{-1}$). After the sample temperature stabilized, propene was inserted at a flow rate of 1.0 cm$^3$ min$^{-1}$ together with He (49.0 cm$^3$ min$^{-1}$) to see the propene adsorption. After 100 min of propene exposure, the sample was purged with pure He for 75 min and subsequently exposed to a flow consisting of NO (5.00 cm$^3$ min$^{-1}$), $O_2$ (10.0 cm$^3$ min$^{-1}$), and He gas (35 cm$^3$ min$^{-1}$) to identify the products of $NO_x$ deoxygenation in the presence of propene.

**Cross-correlation analysis.** We calculated the cross-correlation coefficient of the projected displacements and amplitudes of the reconstructions with 0.5 s time resolution to provide better sensitivity to changes in the data series and to find when displacement and amplitude evolve during the processes (i) and (ii). The Pearson correlation coefficient was calculated for each matrix element in two-dimensional

images as 128 × 128 matrices. The coefficient $\rho(A,B)$ is defined by

$$\rho(A, B) = \frac{1}{N-1} \sum_{i=1}^{N} \left( \frac{A_i - \bar{A}}{\sigma_A} \right) \left( \frac{B_i - \bar{B}}{\sigma_B} \right), \tag{2}$$

where $A$ and $B$ indicate distinct patterns at fixed time points, $N$ is the total number of pixels in the image. $A_i$ and $B_i$ are the intensity value of the $i$th pixel, $\bar{A}$ and $\bar{B}$ are the mean values of the intensities for $A$ and $B$, respectively, $\sigma_A$ and $\sigma_B$ are the standard deviations of the total intensities for $A$ and $B$. Subsequently, the correlation coefficients of 128 × 128 matrices were converted into a single coefficient by averaging all components of the matrix.

The cross-correlation coefficients from the displacement field are shown in Fig. 2g, h for the propene adsorption and the $NO_x$ deoxygenation process, respectively. Supplementary Fig. 4a, b show the cross-correlation from amplitudes for each case. The displacement correlation shows evidence of crossovers over the time series. However, the crossovers in amplitude correlation are not that distinct. This may be because the reconstructed amplitude is proportional to the diffracting electron density and is relatively insensitive to the details of the chemical conditions.

**Powder X-ray diffraction.** X-ray powder diffraction measurements of ZSM-5 zeolite samples were carried out before and after the ion exchange with $CuCl_2$ salt to investigate whether this process affects the zeolite crystal structure. Supplementary Figure 7 shows the powder diffraction results before and after the Cu ion exchange, at RT (a) and 250 °C (b) shown in Supplementary Table 4. The results show good agreement with the previous studies[10,40,41].

**X-ray photoelectron spectroscopy.** We measured the X-ray photoelectron spectroscopy of Cu-ZSM-5 after the ion exchange with $CuCl_2$ salt to investigate whether the chloride exists as a residue. The results are shown in Supplementary Fig. 8, with the peak assignment of each element. From their normalized peak area, we can estimate the ratio of the atomic concentrations[42–44]. Si/Al molar ratio is estimated to 14.99 and Cu/Al molar ratio is 0.6318. Since any signal related to chlorine (198–202 eV) is not observed, this confirms that there are no chloride residuals after the ion exchange with $CuCl_2$.

**X-ray fluorescence microscopy.** XFM measurements were carried out at the 3-ID (Hard X-ray Nanoprobe, HXN) beamline[32,45] at the National Synchrotron Light Source II (NSLS-II, Brookhaven National Laboratory, USA). Cu-ZSM-5 zeolite samples were prepared on a standard sample mounting chip comprising 10 μm thick [100] silicon membrane, 0.5 × 1.4 mm$^2$ in size. The as-synthesized ZSM-5 microcrystals were attached to the silicon membrane using 0.035 wt.% PEI as an adhesive and then calcined at 550 °C for 12 h to remove residual organics. Then the Cu ions were exchanged as described above. The measurement was conducted with 12 keV incident coherent X-rays using the multilayer Laue lens setup. A 1.2 × 1.2 μm$^2$ area was scanned with 5 nm steps and 0.1 s dwell time. The fluorescence signal was collected by a Vortex-ME3 3-element silicon drift detector. The Cu-K map was obtained from the spectrum fitting at each pixel.

**Finite element analysis.** FEA using the COMSOL MULTIPHYSICS package was employed to simulate the deformation field distribution inside the Cu-ZSM-5 crystals for two cases. We made a model with regions of different Cu ion density of Cu-ZSM-5 based on the XFM image. In the simulation in Figs. 3 and 4, the size of the Cu-ZSM-5 crystal is set to $1.88 \times 0.78 \times 2.68 \ \mu\text{m}^3$ on Si substrate. In Fig. 3, we used the strain-rate coefficients of Cu-ZSM-5, $\alpha(t)_{(200)} = 0.3364 \times 10^{-6} \ \text{s}^{-1}$, $\alpha(t)_{(020)} = 0.1275 \times 10^{-6} \ \text{s}^{-1}$ for the non-absorbed area and $\alpha(t)_{(200)} = -0.4878 \times 10^{-6} \ \text{s}^{-1}$, $\alpha(t)_{(020)} = -0.5046 \times 10^{-6} \ \text{s}^{-1}$ for the high Cu density regions (Supplementary Table 2). The coefficients for Si substrate were fixed to be 0 s$^{-1}$. A density of 2250 kg m$^{-3}$, Young's modulus of 79.6 GPa, and Poisson's ratio of 0.26 for pristine ZSM-5 at room temperature were used in the simulation. To compare with the projected displacements from CXD results, the displacement in the $xz$-plane was added those calculated along the $y$-direction, i.e., the (020) direction. The accumulation range was the full size of the crystal in the $y$-direction, $-0.39$ to $+0.39$ μm, and the number of $xz$-plane slices was 1000.

In Fig. 4d, h, we used the stationary linear elastic model. The external stress tensor $\sigma_{\text{far}}$ and $\sigma_{\text{close}}$ obtained from the DFT calculation are directly applied to the high Cu density regions.

**Density functional theory calculations.** The first-principles calculations of structures and electronic properties were conducted using the CASTEP code, which is a plane-wave, pseudopotential program based on DFT, implemented with the BIOVIA Material Studio 6.1 package[46]. Because CASTEP is based on the supercell method, a 3D periodic model of the unit cell was constructed and used for all calculations. Because the entire structure of zeolite shares their electric charges, two Al atoms were randomly replacing the Si positions in ZSM-5 pores. One Cu(II) ion was initially positioned around an Al site at the pore having the largest opening. To calculate the most stable configurations of propene, NO, and $O_2$ adsorbed on Cu-ZSM-5, the conditions were based on the FTIR (Supplementary Fig. 2) and mass spectroscopy (Supplementary Fig. 3) results. One model was Cu(I)-ZSM-5

($Cu(I)$-$Si_{94}Al_2O_{192}$) with one propene, four $O_2$ and one NO, assuming that the propene molecule is near the $Cu(I)$ ion, but NO and $O_2$ molecules were separated in another channel ("far"). Another model assumed that all molecules were near a $Cu(I)$ ion within the same pore ("close"). With these two models, we conducted geometry optimization processes with no constraints, using local density approximation (LDA) with the CA-PZ functional. Ion-electron interactions were modeled in reciprocal space, and the ultrasoft pseudopotential and Koelling-Hamon relativistic treatment were employed. The plane wave cutoff energy was set to 351 eV, the self-consistent field (SCF) convergence threshold was $2.0 \times 10^{-6}$ eV atom$^{-1}$, and k-point sampling with a Monkhorst−Pack grid having 0.5 nm$^{-1}$ k-point spacing was applied. The convergence tolerances for energy, force, stress, and displacement were $2.0 \times 10^{-5}$ eV atom$^{-1}$, 0.5 eV nm$^{-1}$, 0.1 GPa and 0.0002 nm, respectively. The most stable structures for the two models that satisfied the optimization conditions were obtained and shown in Fig. 4a–c, e–g.

Using the optimized potentials derived from the DFT calculations, we obtained that the total stress tensor, $\sigma$, for Fig. 4a–c:

$$\sigma_{\text{far}} = \begin{pmatrix} \sigma_{xx} & \sigma_{xy} & \sigma_{xz} \\ \sigma_{yx} & \sigma_{yy} & \sigma_{yz} \\ \sigma_{zx} & \sigma_{zy} & \sigma_{zz} \end{pmatrix} = \begin{pmatrix} -17.65 & -1.255 & 8.734 \\ -1.255 & -6.498 & 8.308 \\ 8.734 & 8.308 & -5.906 \end{pmatrix}, \quad (3)$$

and that for Fig. 4e–g:

$$\sigma_{\text{close}} = \begin{pmatrix} \sigma_{xx} & \sigma_{xy} & \sigma_{xz} \\ \sigma_{yx} & \sigma_{yy} & \sigma_{yz} \\ \sigma_{zx} & \sigma_{zy} & \sigma_{zz} \end{pmatrix} = \begin{pmatrix} -1.857 & 9.626 & -9.087 \\ 9.626 & 18.80 & -0.903 \\ -9.087 & -0.903 & -4.857 \end{pmatrix}. \quad (4)$$

The values are in the unit of $10^6$ N m$^{-2}$.

**Measurements in chemically deactivated conditions.** To clarify the origin of the deformation observed, the CXD patterns of same crystal were also measured under chemically deactivated conditions, i.e., in $N_2$ as presented in Supplementary Fig. 10a and in NO and $O_2$ without propene adsorbed in Supplementary Fig. 10b. The zeolite crystal in $N_2$ condition shows the same unstrained CXD patterns as in the 'initial' condition of Fig. 2a at $t = 0$ s.

Without hydrocarbons, the catalytic activation temperature for $NO_x$ deoxygenation is >500 °C. Therefore we did not observe any changes with only NO and $O_2$ until 700 s.

## Data availability

The data reported in this paper are available from the corresponding author upon reasonable request.

## Code availability

The code reported in this paper, including the reconstruction algorithm, is available from the corresponding author upon reasonable request.

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

## Acknowledgements

This research was supported by Basic Science Research Program through the National Research Foundation of Korea (NRF-2014R1A2A1A10052454, 2015R1A5A1009962, and 2019R1A6B2A02100883). This research used the HXN Beamline of the National Synchrotron Light Source II, a U.S. Department of Energy (DOE) Office of Science User Facility operated for the DOE Office of Science by Brookhaven National Laboratory under Contract No. DE-SC0012704. Use of the Linac Coherent Light Source (LCLS), SLAC National Accelerator Laboratory, is supported by the U.S. Department of Energy, Office of Science, Office of Basic Energy Sciences under Contract No. DE-AC02-76SF00515.

## Author contributions

H.K. supervised and coordinated all aspects of the project. ZSM-5 growth was carried out by N.H.T. under the supervision of K.B.Y.. Coherent X-ray diffraction measurements were carried out by J.K., J.C., M.C., D.K., G.A.,W.C., R.H., S.S., M.S., A.R., I.K.R., and H.K. CXD data analysis was carried out by J.K., M.C. and D.K. with analysis code by K.Y. and J.N.C. Infrared spectroscopy measurements were done by J.K., H.L., and Y.N.C. Gas analysis measurements were carried out by J.K. and H.L. M.K.S. confirmed theoretically the catalytic process and calculated density functional theory. Scanning Electron Microscopy, X-ray diffraction, X-ray photoelectron spectroscopy measurements were done by N.H.T. and J.K. X-ray fluorescence microscopy measurements were performed by X.H. and Y.C. Finite element analysis calculation was carried out J.K. and Jae. K. under the supervision of H.K. J.K., A.R., K.B.Y., I.K.R., and H.K. wrote the paper. All authors discussed the results and commented on the paper.

## Competing interests

The authors declare no competing interests.
