## [Peer Review File · Nature Communications]

Reviewers' comments:

Reviewer #1 (Remarks to the Author):

I have now read the revised paper, along with the referee reports, and rebuttal letter. It is obvious that the authors did not really have addressed the main problem that the paper does not reveal too much new chemistry and that is e.g. reflected by the request to work out the SAPO-34/SSZ-13 crystal system (among other things to really make it a chemistry/materials story). When looking on the crystals shown in Figure A it is clear that this is not a very good system; and shows that the authors do not have the experience but also did not reach out to people to help them to make such materials. You can have beautiful crystals, both for the CHA as the MFI framework structure. This is just an example, but also for the questions of the other referee (3) the authors are not able to address his/her comments. Summarizing, this revised article is not yet up to the required standard; hence I cannot recommend it for publication in this journal.

Reviewer #2 (Remarks to the Author):

Report on manuscript NCOMMS-20-06598-T entitled 'Time-resolved in situ visualization of the structural response of zeolites during catalysis'

I have been asked to report on the above manuscript in place of Referee #2 who could not attend the whole review process. I got to know this manuscript during the second round. While I have been asked to address solely the question raised by Referee #2 during the first round, I think it is my duty as a referee to make sure that the manuscript is scientifically sound to the best of my understanding. Therefore, I will address not only questions raised by Referee #2 but also some additional technical points, which, I believe, deserve a closer look.

Questions raised by referee #2 and answer brought by the authors:

Most of the points raised by referee #2 have been answered except the question on the resolution. The estimation of the resolution is mentioned for the 3D data, presented in the supplementary information. However, the given value (about 47 nm) does take into account the anisotropy of the resolution.

For the 2D maps presented in the main text, I agree with Referee #2 that the spatial resolution is not extremely high, probably around 200 nm along z and 100 nm along x (according to the number of fringes in the diffraction patterns). This should be mentioned. Furthermore, when comparing FEA results to the experimental ones, one should consider taking the resolution effect into account, in order to provide a fair comparison with the experimental results.

Additional technical questions

1 – I think there is a (hopefully little) problem in the way the phase integration is done.

As a start, the diffraction geometry is a bit confusing when considering Fig.1 (it looks that the beam is entering from the (xz) plane and the exit beam is not clear). Adding the different axis of the crystal should help.

However, given the information in the text and in the inset of Figure 1, one can understand that the exit direction is at about 3 degree from the horizontal plane (the Bragg angle). It is this direction, which has to be considered for the projection of the effective electron density (eq. given on lines 126-127 of the supplementary information) and not y. I agree that 3° is very small and should not play any significant role but this is an assumption which needs to be clearly mentioned. Once these clarified, the main problem comes from the way this integration is done. The quantity to project (or sum up) is the effective electron density, which contains both amplitude and phase. The integration has to be done considering the sum of the effective electron density along the integration direction. In the FEA model (detailed in supplementary Fig. 5), the presence of Cu is not invariant along the direction of integration. Therefore, one can NOT write that $\langle \rho \exp(i\phi) \rangle = \langle \rho \rangle \cdot \exp(i\langle \phi \rangle)$.

(as an exemple, consider the two following electron densities: $\exp(i\pi/2)$ and 2. The mean of their

phase is $\pi/4$, while the mean of the effective electron density has a phase clearly smaller than $\pi/2$).

Depending on the electron contrast and the density between the Cu rich regions and the other parts of the crystal, this may affect the sum. At least, a significant amount of Cu is expected to induce a displacement field over large distance. This needs to be re-evaluated and discussed, if needed. Furthermore, all displacement evaluations shown in the 2D projected plane should be done according to the left term $\langle \rho \exp(i\phi) \rangle$.

2 - The longitudinal length is likely smaller than the sample thickness. It should lead to some phase distributions in the retrieved image. A mean of comparing this effect to the one induced by the two considered processes is to look at the phase obtained in the crystal before the catalysis process. This is done in 3D, with a color scale, which is a bit confusing with this respect as it flattens all small phase fluctuations. Furthermore, in 2D, the retrieved results from the expression given above. It would be straightforward to extract, by inverting the central 2D slice obtained in the 3D diffraction pattern and should be presented, as a starting point.

3 - Regarding Fig. 3f (experimental) and g (model), I am puzzled by the fact the those maps, which corresponds to the differences between the displacements obtained at 250.5 s and 251 s present similar contrasts (even more considering that g is more contrasted than f): Indeed, the model maps in d do not show any visible difference on the chosen color scale, while the experimental maps present a well visible difference, on the same color scale. Furthermore, the color scale chosen to represent both f and g has been truncated and distorted, so that any variation between about 0.5 and -4.5 appear green-bluish. I suspect some mistake here and I would advise to careful check the found values and plot the difference maps using a well-contrasted color scale. Finally one may ask why the change between 250.5 and 251 s is so abrupt while the other propene adsorption steps seem to be smoother.

Authors' Response to Reviewer #1:

I have now read the revised paper, along with the referee reports, and rebuttal letter. It is obvious that the authors did not really have addressed the main problem that the paper does not reveal too much new chemistry and that is e.g. reflected by the request to work out the SAPO-34/SSZ-13 crystal system (among other things to really make it a chemistry/materials story). When looking on the crystals shown in Figure A it is clear that this is not a very good system; and shows that the authors do not have the experience but also did not reach out to people to help them to make such materials. You can have beautiful crystals, both for the CHA as the MFI framework structure. This is just an example, but also for the questions of the other referee (3) the authors are not able to address his/her comments. Summarizing, this revised article is not yet up to the required standard; hence I cannot recommend it for publication in this journal.

Response: Our work is to understand the structural response during the catalytic process, not to find better catalysts or to find new chemistry. Therefore, we select ZSM-5 as an example to show the adsorption and catalytic effect on the strain of the crystal. We have looked at both commercial SAPO and lab-made SSZ-13 chabazite materials with our BCDI coherent diffraction methods and found the crystallinity to be so poor that they are totally unsuitable for the detailed strain analysis we are presenting for ZSM-5. (See Xue Bai, et al, RSC Advances 8 33631 (2018))

Authors' Response to Reviewer #2:

Report on manuscript NCOMMS-20-06598-T entitled 'Time-resolved in situ visualization of the structural response of zeolites during catalysis'

I have been asked to report on the above manuscript in place of Referee #2 who could not attend the whole review process. I got to know this manuscript during the second round. While I have been asked to address solely the question raised by Referee #2 during the first round, I think it is my duty as a referee to make sure that the manuscript is scientifically sound to the best of my understanding. Therefore, I will address not only questions raised by Referee #2 but also some additional technical points, which, I believe, deserve a closer look.

Questions raised by referee #2 and answer brought by the authors:

Most of the points raised by referee #2 have been answered except the question on the resolution. The estimation of the resolution is mentioned for the 3D data, presented in the supplementary information. However, the given value (about 47 nm) does not take into account the anisotropy of the resolution.

For the 2D maps presented in the main text, I agree with Referee #2 that the spatial resolution is not extremely high, probably around 200 nm along z and 100 nm along x (according to the number of fringes in the diffraction patterns). This should be mentioned. Furthermore, when comparing FEA results to the experimental ones, one should consider taking the resolution effect into account, in order to provide a fair comparison with the experimental results.

Response: We estimated the spatial resolution is 72.5 nm along x and 72.5 nm along z by the phase retrieval transfer function calculation. We added this information to the Methods section in the main text (line 237~238).

Additional technical questions

1 - I think there is a (hopefully little) problem in the way the phase integration is done.

As a start, the diffraction geometry is a bit confusing when considering Fig.1 (it looks that the beam is entering from the (xz) plane and the exit beam is not clear). Adding the different axis of the crystal should help. However, given the information in the text and in the inset of Figure 1, one can understand that the exit direction is at about 3 degree from the horizontal plane (the Bragg angle). It is this direction, which has to be considered for the projection of the effective electron density (eq. given on lines 126-127 of the supplementary information) and not y. I agree that 3° is very small and should not play any significant role but this is an assumption which needs to be clearly mentioned.

Response: We thank the referee for the suggestion. We modified Figure 1 below to show the exit direction clearly and added the q-axis indication on the 2D diffraction patterns. It would be helpful to figure out the exit beam direction and the plane of the diffraction patterns taken.

With this geometry, we stated already the theoretical condition $((\mathbf{q} - \mathbf{G}) \cdot \mathbf{u} \ll 2\pi)$ on line 125 in the Supplementary Information. We added the following sentence on the line 134~139 in SI.

“In the experiment, the exit angle of X-ray beam is 3.624° , corresponding to $|\mathbf{q}| = 0.563 \text{ \AA}^{-1}$. The maximum displacement $|\mathbf{u}| \sim 1.1 \text{ \AA}$, which corresponds to $\sim 0.22 \pi$ of (200) lattice spacing of ZSM-5. It satisfies $(\mathbf{q} - \mathbf{G}) \cdot \mathbf{u} \ll 2\pi$. Therefore, the amplitude and the phase from the 2-dimensional image reconstruction are interpreted as the sum of the electron density and the average of displacement along y direction, respectively.”

Once these clarified, the main problem comes from the way this integration is done. The quantity to project (or sum up) is the effective electron density, which contains both amplitude and phase. The integration has to be done considering the sum of the effective electron density along the integration direction. In the FEA model (detailed in supplementary Fig. 5), the presence of Cu is not invariant along the direction of integration. Therefore, one can NOT write that $\langle \rho \exp(i\phi) \rangle = \langle \rho \rangle \cdot \exp(i)$.

(as an exemple, consider the two following electron densities: $\exp(i\pi/2)$ and 2. The mean of their phase is $\pi/4$, while the mean of the effective electron density has a phase clearly smaller than $\pi/2$).

Depending on the electron contrast and the density between the Cu rich regions and the other parts of

the crystal, this may affect the sum. At least, a significant amount of Cu is expected to induce a displacement field over large distance. This needs to be re-evaluated and discussed, if needed. Furthermore, all displacement evaluations shown in the 2D projected plane should be done according to the left term $\langle \rho \exp(i\phi) \rangle$.

Response: The expression mentioned above is from the calculation of the Bragg peak intensity $I(\mathbf{q})$. It is known that the Cu ions by the ion exchange process are not substitutional replacements of the compositional atoms in ZSM-5 and placed near the Al sites. Besides, the Cu ions do not have a fixed distance from the Al atoms. Therefore the Cu ions are inhomogeneously placed inside of the ZSM-5 channels, and the Cu ions cannot directly contribute to the Bragg intensity. Thus we can write the term $I(\mathbf{q})$ as we described in the Supplementary Information.

2 - The longitudinal length is likely smaller than the sample thickness. It should lead to some phase distributions in the retrieved image. A mean of comparing this effect to the one induced by the two considered processes is to look at the phase obtained in the crystal before the catalysis process. This is done in 3D, with a color scale, which is a bit confusing with this respect as it flattens all small phase fluctuations. Furthermore, in 2D, the retrieved results from the expression given above. It would be straightforward to extract, by inverting the central 2D slice obtained in the 3D diffraction pattern and should be presented, as a starting point.

Response: We used the monochromatic beam at $E = 8.800$ keV via Si (1 1 1) monochromator in the experiment. The bandpass is $\Delta\lambda/\lambda = 1.36 \times 10^{-4}$, which corresponds with the longitudinal coherence length of $1.1 \mu\text{m}$ (See Lee, S., et al., Opt. Express 20, 9790 (2012)). In this experiment, the sample thickness is $0.60\text{-}0.80 \mu\text{m}$. Thus the longitudinal coherence length is sufficient for our sample.

In addition, our original 3D imaging results in Fig. S1 was generated from the accumulated coherent diffraction patterns by 2400 shots. We replaced the 3D image with 60 shots, which have the same S/N ratio with 2D images. We added a description of the 3D imaging data (line 216~218, in p.10).

3 - Regarding Fig. 3f (experimental) and g (model), I am puzzled by the fact the those maps, which corresponds to the differences between the displacements obtained at 250.5 s and 251 s present similar contrasts (even more considering that g is more contrasted than f): Indeed, the model maps in d do not show any visible difference on the chosen color scale, while the experimental maps present a well visible difference, on the same color scale. Furthermore, the color scale chosen to represent both f and g has been truncated and distorted, so that any variation between about 0.5 and -4.5 appear green-bluish. I suspect some mistake here and I would advise to careful check the found values and plot the difference maps using a well-contrasted color scale. Finally one may ask why the change between 250.5 and 251 s is so abrupt while the other propene adsorption steps seem to be smoother.

Response: It was mistakenly applied to a different scale. We changed the color bar of Figs. 3f and 3g in the same scale.

REVIEWER COMMENTS

Reviewer #1 (Remarks to the Author):

Second report on manuscript NCOMMS-20-06598-T entitled 'Time-resolved in situ visualization of the structural response of zeolites during catalysis'

I am refereeing this manuscript for the second time and while I can see that some efforts have been made to improve the manuscript, I am not fully satisfied with most of the answers given by the authors. As far as I can see, my questions and comments do not point out to major mistakes, which would invalidate the found results. However, the series of inaccuracies, which I have highlighted in my previous report and which are still present in the revised manuscript, makes the whole work fragile and poorly justified, clearly below the scientific standards of Nature Communications. In these circumstances, I can not recommend publication.

1 - The resolution: there is no evidence regarding the values found for the spatial resolution. In addition, I anticipate that the resolution is anisotropic, as a result of the presence of the sharp edge. This was not discussed.

2a - Integration direction: The integration direction should be the exit beam direction and not y . This mistake comes from the fact that k is taken as 0 in the calculation of p.7 supplementary information. In the detection plane, $k = 0$ is only true at the Bragg peak. As the detector is slightly inclined with respect to G , k is not strictly 0. This is not a big issue, but should be justified properly.

2a - Integration method: I still think that one can not write $\langle \rho \exp(i\phi) \rangle = \langle \rho \rangle \cdot \exp(i\langle \phi \rangle)$ (1) as used by the authors.

In the revised manuscript, the authors justify this approximation (p. 7 SI) by invoking $(q - G) \cdot u < 2\pi$. I do not see how this inequality can help to transform the left hand term of Eq. (1) above, into the right hand term.

In the rebuttal letter, the authors are invoking the position of the Cu Ion in the cell to justify the approximation. I do not understand the link. On the contrary, the Cu ions are strongly distributed in the crystal, reinforcing the need to confirm the possibility to use the approximation in (1).

3 - Longitudinal coherence length versus the thickness of the sample. The answer is not satisfying. The Bragg vector being along x , it is the thickness along x , which has to be compared to the projection of the longitudinal coherent length. In this direction, the crystal is about 2 microns. The analysis of the parasitic effects (including the partial coherence effects) prior to propene adsorption or NO_x deoxygenation analysis in 2D is not done. Therefore, it is not clear what is the origin of the initial displacement state.

4 - About the temporal analysis of Fig. 3: the question regarding the plots f and g has been taken into account.

About the temporal resolution, there is an abrupt change between 250.5 and 251 s, which is not really commented: could the authors plot the displacement as a function of time, in the region which shows an abrupt change, to see whether it is significant for the whole data set ?

5 - Additional comments: reading through the manuscript again, I found very puzzling the mathematical derivation shown in p. 3 of the supplementary information. Indeed $g'_m(x)$ is a function of $\theta'(k)$. Same comment for $g_{m+1}(x)$; several variables are not defined (e.g., g' , η , ϕ). This deserves a closer look.

AUTHOR'S RESPONSE TO REVIEWER 1:

Second report on manuscript NCOMMS-20-06598-T entitled 'Time-resolved in situ visualization of the structural response of zeolites during catalysis'

I am refereeing this manuscript for the second time and while I can see that some efforts have been made to improve the manuscript, I am not fully satisfied with most of the answers given by the authors. As far as I can see, my questions and comments do not point out to major mistakes, which would invalidate the found results. However, the series of inaccuracies, which I have highlighted in my previous report and which are still present in the revised manuscript, makes the whole work fragile and poorly justified, clearly below the scientific standards of Nature Communications. In these circumstances, I can not recommend publication.

1 - The resolution: there is no evidence regarding the values found for the spatial resolution. In addition, I anticipate that the resolution is anisotropic, as a result of the presence of the sharp edge. This was not discussed.

Response: As we explained in the previous rebuttal letter, the spatial resolutions in x and z direction were calculated by phase retrieval transfer function (PRTF). Now along with the referee's suggestion, the detailed description and the following results below are added in the SI.

Assuming that a reconstructed image in real space, $g(\mathbf{x})$, is obtained by the phase retrieval process, its Fourier transform is expressed by $\mathcal{F}_{\mathbf{w}}(g(\mathbf{x})) = G(\mathbf{w}) = |G| \exp(i\varphi(\mathbf{w}))$, where $\varphi(\mathbf{w})$ is the retrieved phase. \mathbf{w} is the spatial frequency, in which each spatial frequency can be regarded as a volume grating. The square of the modulus, $|G|^2$, is the measured intensity $I(\mathbf{w})$. PRTF is defined by

$$PRTF(\mathbf{w}) = \frac{|G(\mathbf{w})|}{\sqrt{I(\mathbf{w})}} = \frac{||G| \exp(i\varphi(\mathbf{w}))|}{\sqrt{I(\mathbf{w})}}.$$

The resolution cutoff of the phase retrieval process is estimated conservatively as the spatial frequency \mathbf{w} when the PRTF reaches 0.5 (Chapman, H. N. et al., *J. Opt. Soc. Am. A* **23**, 1179-1200 (2006)). We show the PRTF as a function of w_z and w_x in the Figures below. We used ~4700 data sets for the error bars in the resolution of the 2D reconstruction. The respective resolution cutoff in z and x direction, denoted as l_z and l_x , respectively, are 74.9 ± 1.1 nm. We added this information on page 10 of the SI in addition to the value of the 3D.

2a - Integration direction: The integration direction should be the exit beam direction and not y. This mistake comes from the fact that k is taken as 0 in the calculation of p.7 supplementary information. In the detection plane, k = 0 is only true at the Bragg peak. As the detector is slightly inclined with respect to G, k is not strictly 0. This is not a big issue, but should be justified properly.

Response: There might be some confusion in notation. The reason we take k = 0 is that we are considering a 2D slice in the xz plane (see Ref. 11 in the SI). We speculate that “k” in the referee’s comment means the scattering wavevector (e.g., k_{out}).

As the referee mentioned, the integration direction should be the exit beam direction. However, in this study, the angle between the exit beam direction and the y direction is only $\sim 3.6^\circ$. Assuming that the thickness in the y direction is $1.0 \mu\text{m}$, a path length difference is $0.00197 \mu\text{m}$, i.e. $\sim 0.2 \%$ of the sample thickness. We added this discussion in SI (page 8, line 20~24).

2a - Integration method: I still think that one cannot write $\langle \rho \exp(i\phi) \rangle = \langle \rho \rangle \cdot \exp(i)$ (1) as used by the authors.

In the revised manuscript, the authors justify this approximation (p. 7 SI) by invoking $(q - G) \cdot u \ll 2\pi$. I do not see how this inequality can help to transform the left hand term of Eq. (1) above, into the right hand term.

In the rebuttal letter, the authors are invoking the position of the Cu Ion in the cell to justify the approximation. I do not understand the link. On the contrary, the Cu ions are strongly distributed in the crystal, reinforcing the need to confirm the possibility to use the approximation in (1).

Response: Thank you very much for the comments. We rewrote the following in the SI (p. 6-8).

In our case, the Cu ions are introduced in the ZSM-5 microcrystals by ion exchange process and the ion exchange process does not substitute Cu ion for Al atoms. Therefore the Cu ion cannot directly affect the structure factor $s(x, y, z, \mathbf{G})$.

However, the displacement field $\mathbf{u}(x, y, z)$ changes during the catalytic process of Cu ion. Then we can write $\mathbf{u}(x, y, z)$ as $\mathbf{u}(x, y, z) + \Delta\mathbf{u}(x, y, z)$. Then $\tilde{\rho}(x, z)$ can be written as

$$\tilde{\rho}(x, z) = \sum_y s(x, y, z, \mathbf{G}) \exp\left(i\mathbf{G} \cdot \mathbf{u}(x, y, z) \cdot (1 + \beta(x, y, z))\right)$$

where β is defined by the deformation coefficient, which is the strain-rate coefficient multiplied by the measurement time t.

$$\beta(x, y, z) = \frac{\Delta\mathbf{u}(x, y, z)}{\mathbf{u}(x, y, z)} = \alpha(x, y, z, t) \cdot t$$

For the theoretical values, the structure factor $s(x, y, z, \mathbf{G})$ and displacement field $\mathbf{u}(x, y, z) \cdot (1 + \beta(x, y, z))$ are calculated for each element $(\Delta x, \Delta y, \Delta z)$ at (x, y, z) and sum over the y direction.

The electron density might be changed due to $\Delta \mathbf{u}$ (even if it is very small) and $\Delta \mathbf{u}$ is varied at different y . Therefore, we cannot take the $s(x, y, z, \mathbf{q})$ out of the summation Σ_y .

The model we used in this study is that $s(x, y, z, \mathbf{G})$ has the form of a step function-like in x , y and z direction. Referring to Supplementary Fig. 7, Supplementary Tables 2 and 3, propene non-absorbed regions have a constant structure factor C_{vac} and the propene absorbed regions have also other constants $C_{\text{prop},n}$ ($n = 1, 2, 3, \dots$) with different order (n) of the adsorption of propene to match $\alpha(x, y, z, t)$ and $\beta(x, y, z)$ for specific positions. Thus the actual integration process in y direction is made in a simple form:

$$s(x, y, z, \mathbf{G}) = \begin{cases} C_{\text{vac}} & \{x, y, z\} \in A_{\text{vac}} \\ C_{\text{prop},1} & \{x, y, z\} \in A_{\text{prop},1} \\ C_{\text{prop},2} & \{x, y, z\} \in A_{\text{prop},2} \\ \vdots & \vdots \end{cases}$$

where A_{vac} and $A_{\text{prop},n}$ denote the regions (shown in Supplementary Fig. 7) of propene non-absorbed and propene absorbed, respectively. Then the $\tilde{\rho}(x, z)$ becomes

$$\begin{aligned} \tilde{\rho}(x, z) = & \sum_{y \in A_{\text{vac}}} C_{\text{vac}} \cdot \exp\left(i\mathbf{G} \cdot \mathbf{u}(x, y, z) \cdot (1 + \beta(x, y, z))\right) \\ & + \sum_{y \in A_{\text{prop},1}} C_{\text{prop},1} \cdot \exp\left(i\mathbf{G} \cdot \mathbf{u}(x, y, z) \cdot (1 + \beta(x, y, z))\right) \\ & + \sum_{y \in A_{\text{prop},2}} C_{\text{prop},2} \cdot \exp\left(i\mathbf{G} \cdot \mathbf{u}(x, y, z) \cdot (1 + \beta(x, y, z))\right) + \dots \end{aligned}$$

Therefore, as the answer of the referee's question, $\langle \exp(i \cdot \phi) \rangle = \langle \rho \rangle \cdot \exp(i)$ is now valid for each of the individual sums.

On the other hand, we could consider the extreme case of the Cu ion's effect in the structure factor by assuming a complex of the Cu ion-Al atom with a specific distance. One Cu ion can be attached to one Al atom. In the calculation of scattering amplitude, the number of Al and Cu is referred from the Si/Al ratio (44) and Cu/Al ratio (0.6318) (See the information in the Methods and X-ray photoelectron spectroscopy results in the SI). The result shows the unit cell structure factor is increased by 0.0956%.

3 - Longitudinal coherence length versus the thickness of the sample. The answer is not satisfying. The Bragg vector being along x , it is the thickness along x , which has to be compared to the projection of the longitudinal coherent length. In this direction, the crystal is about 2 microns.

The analysis of the parasitic effects (including the partial coherence effects) prior to propene adsorption or NOx deoxygenation analysis in 2D is not done. Therefore, it is not clear what is the origin of the initial displacement state.

Response: The longitudinal coherence length must be related to the path length along the beam propagation; y in this case, not the Bragg vector. “ x ” direction is related to the coherence length in the transverse direction, not longitudinal.

If we calculate the path length through the sample regarding the incident and exit angles, it is $0.601 \sim 0.802 \mu\text{m}$, still smaller than the longitudinal coherence length $L_\lambda = 1.1 \mu\text{m}$ (Lee, S., et al., *Opt. Express* **20**, 9790-9800 (2012)).

For the sample size in x and z direction, we have to consider the transverse coherence lengths. Since the LCLS FEL beam is $\sim 93\%$ transversely coherent (Lee, S., et al., *Opt. Express* **21**, 24647-24664 (2013)), the transverse coherence length of the beam is $L_x = L_z \sim 9.3 \times 10^4 \mu\text{m}$, which is larger than the sample dimensions in x and z .

The information about the longitudinal and transverse coherence lengths is added in the Methods section on the manuscript (See line 3 and 7-11, p. 10).

4 - About the temporal analysis of Fig. 3: the question regarding the plots f and g has been taken into account.

About the temporal resolution, there is an abrupt change between 250.5 and 251 s, which is not really commented: could the authors plot the displacement as a function of time, in the region which shows an abrupt change, to see whether it is significant for the whole data set ?

Response: As we discussed in section IV of SI, we determined the number of FEL pulses to average upon for defining optimized conditions between time resolution and the signal-to-noise ratio. The shape of the object, as determined by the reconstructed amplitude, remains persistent independently of the number of shots used so long as an average of 60 or more shots. Therefore we used an averaging of 60 shots corresponding to a 0.5 s time resolution.

At 250.5 and 251.0 s, where $\Delta a/a$ is in the plateau in Fig. 2Ac, there is an alternating displacement field pattern in the form of connected columns in arched shapes. Shown below as well as in Supplementary Fig. 6, the displacement fields in 250.5 ~ 251.0 s and 252.5 ~ 253.0 s range show similar and alternating patterns. We selected the data at 250.5 and 251.0 s for demonstrating the detailed process of adsorption using the strain analysis. It is interpreted that propene molecules are adsorbed at the outer side of the internal ring. We added this in the manuscript (on p. 6).

5 - Additional comments: reading through the manuscript again, I found very puzzling the mathematical derivation shown in p. 3 of the supplementary information. Indeed $g'm(x)$ is a function of $\theta'(k)$. Same comment for $g_{m+1}(x)$; several variables are not defined (e.g., g' , η , ϕ). This deserves a closer look.

Response: Thank you very much for finding a mistake. It was inadvertently mistyped in $\theta'(k)$ instead of $\theta'(x)$. We corrected it in the equations in SI.

$$\begin{aligned}g'_m(\mathbf{x}) &= |g'_m(\mathbf{x})| \exp[i\theta'_m(\mathbf{x})], \\g_{m+1}(\mathbf{x}) &= |f(\mathbf{x})| \exp[i\theta'_{m+1}(\mathbf{x})].\end{aligned}$$